# Contributors to Preterm Birth: Data from a Single Polish Perinatal Center

**DOI:** 10.3390/children10030447

**Published:** 2023-02-25

**Authors:** Iwona Jańczewska, Monika Cichoń-Kotek, Małgorzata Glińska, Katarzyna Deptulska-Hurko, Krzysztof Basiński, Mateusz Woźniak, Marek Wiergowski, Marek Biziuk, Anna Szablewska, Mikołaj Cichoń, Jolanta Wierzba

**Affiliations:** 1Department of Neonatology, Medical University of Gdańsk, Mariana Smoluchowskiego 17 Street, 80-214 Gdańsk, Poland; 2Department of Pediatrics, Hematology and Oncology, Medical University of Gdańsk, Dębinki 7 Street, 80-952 Gdańsk, Poland; 3Division of Quality of Life Research, Medical University of Gdańsk, Tuwima 15 Street, 80-211 Gdańsk, Poland; 4Department of Forensic Medicine, Medical University of Gdańsk, Dębowa 23 Street, 80-211 Gdańsk, Poland; 5Department of Analytical Chemistry, Faculty of Chemistry, Gdańsk University of Technology, Gabriela Narutowicza 11/12 Street, 80-233 Gdańsk, Poland; 6Department of Obstetrical and Gynaecological Nursing, Institute of Nursing and Midwifery, Medical University of Gdańsk, Dębinki 7 Street, 80-214 Gdańsk, Poland; 7Department of Dermatology, Venerology and Allergology, Faculty of Medicine, Medical University of Gdańsk, Mariana Smoluchowskiego 17 Street, 80-211 Gdańsk, Poland; 8Department of Internal and Pediatric Nursing, Institute of Nursing and Midwifery, Medical University of Gdańsk, Dębinki 7 Street, 80-211 Gdańsk, Poland

**Keywords:** preterm birth, risk of preterm birth, pregnancy outcomes

## Abstract

Preterm birth may result from overlapping causes including maternal age, health, previous obstetric history and a variety of social factors. We aimed to identify factors contributing to preterm birth in respect to new social and environmental changes in the reproductive patterns. Our cross-sectional study included 495 mother–infant pairs and was based on maternal self-reporting in an originally developed questionnaire. Neonates were divided into two groups: 72 premature babies (study group) and 423 full-term babies (control group). We analyzed maternal, sociodemographic and economic characteristics, habits, chronic diseases, previous obstetric history and pregnancy complications. For statistical analysis, Pearson’s Chi-squared independence test was used with a statistical significance level of 0.05. Preterm births were more common among mothers living in villages (*p* < 0.001) and with lower education level (*p* = 0.01). Premature births were also positively associated with mothers who were running their own businesses (*p* = 0.031). Mothers with a history of previous miscarriages gave birth at a significantly older age (*p* < 0.001). The most frequent pregnancy complications were hypothyroidism (41.4%), pregestational and gestational diabetes mellitus (DM; 17.8%) and hypertension (8.1%). Pregestational DM significantly influenced the occurrence of prematurity (*p* < 0.05). Pregestational DM, being professionally active, a lower education level and living outside cities are important risk factors of prematurity.

## 1. Introduction

Preterm labor is defined as a birth before 37 completed weeks of gestation (less than 259 days) [1]. Preterm birth is currently one of the most important problems in maternal-child health worldwide and is especially severe in low-income countries. In Europe in 2014, preterm births constituted 8.7% of all births [2]. Countries with the lowest preterm birth rates are Finland and Lithuania (ranging from 5.5–5.9%), whereas the highest rate of preterm births is observed in the Czech Republic (8.1%), Cyprus (10.4%) and Austria (11%). In Poland, the preterm birth rate has increased from 6.0% in 2000 up to 7.0% in 2019 [2,3,4,5]. Despite the introduction of many public health projects and medical interventions to prevent preterm births, even in high-income countries preterm birth is the second most common cause of death after congenital anomalies in neonates and children under five, and it has lifelong effects on neurodevelopmental outcomes in affected individuals. Preterm babies are at increased risk of neurological impairment and disability and chronic disease in adulthood [6]. Multiple factors are associated with preterm birth, including maternal age, race, previous obstetric history, infections, and exposure to cigarettes and drugs. Comorbidities such as obesity, hypertension, hypothyroidism, pregestational and gestational diabetes mellitus (DM) in mothers may contribute to preterm birth [7,8,9,10]. Psychosocial factors that may contribute to preterm birth include marital status, professional activity, living conditions and the level of social security [11]. Genetic, environmental, social and economic factors overlap, and all have remarkable impacts on the health of pregnant women (and the rate of preterm births), especially in developing countries. Notably, in the developed countries the data is not sufficient. Despite the ongoing enhancement of perinatal care in Poland, in recent years we have observed a slightly increasing trend of preterm births. Social norms and significant changes within it have impacts on the lifestyles of individuals. Moreover, numerous women live in informal relationships and/or raise children alone [10,12,13,14]. Bearing in mind that many factors (sociodemographic, economic, lifestyle, work status) can contribute towards the increasing numbers of preterm births in the developed countries, an accurate investigation is needed. Such assessment could be a starting point to improve the health care of pregnant women by implementing new health programs aiming to provide even better health care and education about the aforementioned risk factors of preterm birth, hence decreasing its incidence. Perhaps, some of the risk factors are modifiable and could be easily avoided or changed.

The aim of this preliminary study is to evaluate whether sociodemographic, economic and lifestyle factors as well as work status or chronic diseases contribute to the preterm birth in a cohort of women from the Pomeranian region.

## 2. Materials and Methods

This preliminary study was conducted among mothers who gave birth between February 2019 and February 2020 in the University Clinical Center (UCC) in Gdańsk. Our center is a tertiary perinatal care facility associated with the Medical University of Gdansk (MUG), located in Gdansk, Pomeranian region, north Poland. The Department of Neonatology provides a full range of neonatal care services: intensive, intermediate, continuing care, and well-baby nursery for patients from the entire region. The Obstetrics Department provides the highest level of care for complicated pregnancies and deliveries. During the study, 2800 births took place. This Department provides diagnostic approaches to and treatment of multiple pregnancies, diabetes in pregnancy, birth defects, preterm deliveries, and other adverse pregnancy complications. It also supports physiological deliveries. Maternal and neonatal patients from lower-level hospitals in the region are referred for diagnostics and treatment to the UCC.

The study was based on maternal self-reporting through an original questionnaire in which women described their state of health and any disorders occurring before and during pregnancy, as well as any medications that were prescribed during pregnancy. We analyzed maternal sociodemographic and economic characteristics, maternal habits, previous obstetric history, maternal chronic disease, and pregnancy complications.

The survey was anonymous and in the Polish language. It consisted of 33 closed single- and multiple-choice questions. All questionnaires were gathered in written, “paper form”. Only completely and correctly filled questionaries were enrolled in the statistical analysis. Approximately 10% of them were incomplete and, therefore, were not included in the study. Five percent of patients refused to take part in the survey, mostly due to unsatisfying health conditions postpartum.

In total, we enrolled 495 mother–infant dyads. Neonates were divided into two groups: 72 newborn infants of premature birth (<37 weeks; case group) and 423 newborn infants of term delivery (≥37 weeks; control group). All maternal participants provided their written informed consent. The inclusion criteria were singleton pregnancies, enrolling Polish-speaking women who have just given birth and obtaining a written consent to take part in the study. The exclusion criteria were multiple pregnancies (since they statistically more often lead to preterm birth), inappropriately fulfilled questionnaires and refusal to take part in the study.

This is a preliminary study, which we plan to extend in the future to analyze the spontaneous and iatrogenic causes of preterm birth in a larger group of patients.

### Statistical Analysis

Statistical significance of differences between categorical variables was calculated with Pearson’s Chi-squared independence test. For age and gestational age, means (M) were compared with Welch’s *t*-test, correcting for unequal variances. All analyses were conducted in Python (version 3.8) using packages Pandas (version 1.4) and Pingouin (version 0.5). Statistical significance level was 0.05.

## 3. Results

Fifty percent (50%) of women in our study had a child between the ages of 28 and 35, (M = 31.25 years, SD = 4.93); the youngest woman was 14, and the oldest was 45 years old. The mean age of primiparous was M = 29.4 (SD = 4.7), while the mean age of women who had given birth previously was M = 33 (SD = 4.5).

Of 495 babies, 72 were born prematurely, which is 14.5%. There was no statistically significant difference in the age of mothers giving birth to their babies at term and prematurely (*p* > 0.05).

Nearly half the mothers (47%) were residents of towns with more than 400,000 inhabitants, 31% were residents of smaller towns, and 22% lived in rural areas. There were statistically significant differences in the proportion of premature to full-term neonates depending on the place of residence (*p* < 0.001). The percentage of mothers residing in villages was higher in the group of premature neonates in comparison to the full-term neonates group.

Women with a university education made up 70.5% (349) of the women participating in the study and 83% (409) worked during pregnancy. There were statistically significant differences depending on the mother’s level of education. The rate of prematurity was higher among mothers with lower education levels than among mothers with higher education (*p* = 0.01). We also found the relationship between prematurity and professional activity of women. Proportionally women running their own business were more frequent in the group of preterm birth than in the group of women giving birth at term (*p* = 0.031).

Married and cohabiting mothers constituted 74.3% (368) and 15% (74) of the study group, respectively. There was a correlation between marital status and gestational age at birth: in the group of full-term infants there were more cohabiting than married mothers, while in the premature group married women were more frequent. Most of our respondents declared their socioeconomic status to be either good or very good. This factor did not affect the week of delivery (*p* > 0.05). There were no statistically significant effects of smoking during pregnancy on preterm labor (*p* > 0.05). The general characteristics of the study group are given in Table 1.

A previous history of miscarriage was reported in 21.2% (105). Women who had previously had a miscarriage had a significantly higher maternal age (M = 32.9, SD = 4.93) in comparison to women who had not had a miscarriage (M = 30.8, SD = 4.84; *t* = −3.90, *p* < 0.001).

The most frequent pregnancy complications were hypothyroidism in 41.4% of our cohort, gestational diabetes (GDM) in 15.8% and hypertension in 8.1%. There were no statistically significant differences in hypertension developing before and during pregnancy, hypothyroidism, GDM, chronic kidney disease, asthma, allergies, depression, anaemia, or cardiovascular diseases (all *p*’s > 0.05). Before pregnancy, 10 women (2.0%) suffered from diabetes mellitus (DM) and this group gave birth to premature babies significantly more often than women without DM (*p* < 0.05). Of the entire group, 61.4% (304) required some form of prescription medication, the most often being levothyroxine.

Women receiving oral or intravaginal progesterone during pregnancy accounted for 14.5% (45 cases). Of these, 13 had a history of spontaneous abortion, while 32 did not. In three cases there had been spontaneous preterm deliveries in prior pregnancies. In the group of women taking progesterone, the mean gestational age (GA) at birth was significantly lower, on average 37.0 weeks (SD = 2.69), compared to women not taking progesterone (M = 38.7 weeks, SD = 2.2); *t* = 3.65, *p* < 001. Adverse pregnancy outcomes are given in Table 2.

## 4. Discussion

The overall rates of preterm delivery in the recent decades in European countries such as Cyprus, Greece, Germany, and Poland have increased, while in others such as Estonia, Croatia, and Netherlands the overall rates of prematurity have either stabilized or decreased [7,15]. The regionalization of obstetric and neonatal care and transfer of pregnant women at risk of preterm labor to the tertiary care center leads to decreased mortality and morbidity rates among preterm infants. However, our study confirmed that this practice also results in a much higher frequency of preterm deliveries in these settings compared to the lower level of perinatal care hospitals. Our study population was drawn from the third level of reference care, where more complicated pregnancies and deliveries are carried out.

Numerous recently published studies have shown there are new social and environmental risk factors which contribute to an increased risk of preterm birth for certain women. The determinants of change in reproductive patterns can be explained by the cultural, social, and economic changes occurring in societies of developed countries. In European countries, the decision whether to have children is not without controversy. Many women want to balance motherhood with completing their education, getting a job and maintaining a liberated lifestyle [16,17]. This trend may lead to delayed childbearing and lower fertility rates [16,17,18,19]. According to EUROPERISTAT reports, in the European Union (EU) in 2016 the average age of the mother during her first childbirth was 29.0 years, while in 2019, it was 29.4 years. A similar trend was observed in Poland; although the mean age at which women had their first child was still below 30 years, there was a slight increase from 27.2 to 27.6 between 2016 and 2019 [2,20,21]. Data from our study corresponds with these European statistics. The higher mean age of first delivery among our mothers compared to the overall mean age of primipara women in Poland may be related to the fact that nearly half lived in a large city and had higher education levels.

Over past decades, many highly educated and qualified women have entered the workforce and have started their families while they were working [15,16,17,18]. Many women, rather than entering the formal workforce, create a new business in association with their family environment. These women, referred to as “mumpreneurs”, wish to find a work–life balance as a business owner [22]. However, running their own business may be challenging and stressful. Pregnant women may experience fear of failure, job-related anxiety, and discrimination related to their pregnancy [23,24,25].

Stress during pregnancy may have far-reaching implications, including lower GA at birth in women affected. Moreover, business owners tend to work longer hours, which may also contribute to preterm birth [16,24,26,27]. We found a relationship between prematurity and the professional activity of women. We showed that being a business owner positively correlated with prematurity.

An increased risk of preterm birth associated with both cohabitation and single motherhood among women in EU has been reported [9]. However, when extramarital births became more common in a community, marital status ceased to be a risk factor for prematurity [28,29]. Moreover, some authors have reported that only single mothers living alone have a higher prevalence of preterm births [30,31]. We did not find extramarital relationship to be a risk factor for preterm birth in our cohort.

A lower level of education and living in villages have been shown to be an important determinant of preterm births [10,20,21]. Although the health insurance system in Poland is based on principles of equal treatment and access to healthcare services and pregnant women are eligible for free healthcare benefits, women living in rural areas may have less access to health care as compared with women from urban areas, which can increase the incidence of preterm birth among rural women [10,25]. It seems that education does not constitute a direct, independent risk factor of prematurity, but may be indirectly linked to lower levels of health awareness and thus risk-related behavior and lifestyle choices. Smokers were found to be more likely to give birth to preterm babies in several studies [7,31,32,33,34,35]. We also indicated a relationship between low education levels and prematurity, but we did not find any link between prematurity and smoking. Nevertheless, the promotion of smoking abstinence at childbearing age remains important for the health of mothers and their children, as maternal smoking is a well-known risk factor for perinatal complication.

Postponing the age of motherhood is also related to pregnancy complications, reproductive failures such as miscarriages, and preterm birth [18,36,37]. Medical motives were reported as frequent reasons for delayed motherhood [38], and our study confirmed that mothers with a history of previous miscarriages gave birth at an older age. The prevalence of chronic disease in childbearing women has increased dramatically during the past decades from less than 5% in the late 20th century to almost 16% in the early 21st century [39]. Among the disorders, both GDM and DM have become increasingly common and women with these disorders have an increased risk of a range of complications of pregnancy [40,41,42]. The adverse pregnancy outcomes associated with diabetes include preterm birth. Results from our study confirmed previous reports. DM and GDM were the second highest group of chronic diseases among women of our cohort, but only pregestational DM significantly influenced the occurrence of preterm births. Prior studies have also shown that preterm deliveries remain high in women with DM [41,42,43,44]. The limited literature data did not indicate an association between premature delivery and GDM [45]. Our results revealed that there was no link. X It has been shown that satisfactory maternal glycemic control, particularly in the periconceptional period and in the first trimester of pregnancy, is associated with reduced preterm delivery and neonatal morbidity [44,46]. Therefore, effective education of childbearing women suffering from pregestational DM who are planning to become pregnant appears to be crucial to achieve better obstetric and neonatal outcomes.

Other chronic maternal conditions such as hypothyroidism and hypertensive disorders associated with preterm birth were also described as important contributors of prematurity [47,48].

According to published data, overt hypothyroidism occurs in 0.3–0.5% of pregnancies, and subclinical hypothyroidism (SCH) in 2–3%. Hypothyroidism during pregnancy increases the risk of spontaneous abortion, anemia, pregnancy-induced hypertension (PIH), and placental abruption. It is also associated with neonatal complications such as prematurity, LBW, and reduced intelligence in the offspring of these women. Even women with SCH are more likely to have preterm deliveries. Numerous professional associations advise evaluation of thyroid function at the first prenatal clinical appointment to avoid preterm births and other obstetric complications. Polish guidelines suggest testing of pregnant women to identify thyroid dysfunction and recommend the treatment of thyroid disorders, including SCH, during pregnancy [48,49,50,51].

In our cohort, hypothyroidism was the most common disorder and, consequently, levothyroxine was the most frequently used medicine. We found that neither hypothyroidism nor Levothyroxine intake were associated with an increased incidence of preterm labor. This observation leads us to presume that euthyroxinemia in early pregnancy is crucial for maintaining normal placental development and function and to avoid preterm deliveries. The results of our own research suggest that in the cases in which thyroid diseases were diagnosed early and properly treated, it was possible to indirectly prolong the duration of pregnancy.

In our study, hypertension was a common complication of pregnancy. While this is often mentioned as the cause of prematurity [47], we did not find a similar correlation in our cohort. Methyldopa remains the first-line treatment for hypertension in pregnancy.

Significant efforts have been made to avoid preterm births. This includes the prophylactic administration of progesterone in pregnant women with a history of at least one spontaneous preterm delivery, and in pregnant women without this history with a short cervix in ultrasonographic measurement of cervical length in midgestation [52,53,54]. We found that women taking progesterone were more likely to give birth slightly earlier in comparison to mothers without progesterone. It is probable that this therapeutic intervention in women at risk of preterm labor improved obstetric outcomes and the mean duration of pregnancy was extended near term. Otherwise, the consequent mortality and handicap of infants born too soon could be more serious.

### Limitations of the Study

Limitations of the study are not considering the type of delivery (vaginal birth versus cesarean section) as well as and not considering labor from Assisted Reproductive Technology (ART), such as in vitro fertilization (IVF). The goal of the survey was to collect data about the women’s lifestyle and environmental, social and economic conditions. Therefore, apart from certain concomitant diseases (hypertension, diabetes mellitus or hypothyroidism), our study focuses deeply on determining the influence of health behavior of pregnant women on the risk of preterm birth. Since it is a preliminary study, we did not include information on either ART treatments or type of delivery, as these questions were not included in the survey. For the same reason we did not ask whether the pregnancy ended with preterm birth due to iatrogenic reasons (i.e., elective cesarean section). Nonetheless, due to the increasing number of preterm births in developed countries (including Poland), it is crucial to investigate the iatrogenic reasons responsible for them; this we aim to accomplish in the next study.

## 5. Conclusions

Pregnant women in the workforce are exposed to higher levels of stress, which contributes to preterm births. Having a job, especially running your own business, positively correlated with prematurity, while advanced maternal age positively correlated with miscarriages. Pregestational DM is an important risk factor for preterm delivery. Improving the health condition of diabetic mothers and minimizing the adverse outcomes of diabetes on pregnancy should be prioritized.

## Figures and Tables

**Table 1 children-10-00447-t001:** Characteristics of the study population, *n* = 495.

	**Premature Neonates, Gestation < 37 Weeks** **N = 72 (14.5%)**	**Full-Term Neonates, Gestation > 37 Weeks** **N = 423 (85.5%)**	** *p* **
Mother’s age	Mean 31.50 Std 5.54 Min–max: 18–45	Mean 31.20 Std. 4.83 Min–max: 14–45	>0.05
A place of residence			
Village	27 (37.5%)	80 (19%)	<0.001
Town < 100,000	14 (19.4%)	59 (14%)	
Town 100,000–400,000	7 (9.7%)	69 (16.4%)	
Town > 400,000	24 (33.3%)	213 (50.6%)	

Education			
Primary	12 (16.7%)	30 (7.1%)	0.01
Secondary	18 (25.0%)	86 (20.1%)	
Upper secondary	42 (58.3%)	307 (72.7%)	

Occupational status			
Employed	43 (60.6%)	305 (72.6%)	0.031
Own business	16 (22.5%)	45 (10.7%)	
Stay-at-home mother	11 (15.5%)	57 (13.6%)	
Student	1 (1.4%)	13 (3.1%)	

Civil status			
Married	60 (83.3%)	308 (72.8%)	0.023
Cohabiting/Extramarital relationship/	3 (4.2%)	71 (16.8%)	
Divorced/single/with parents	9 12.5%	44 (10.4%)	

Socioeconomic status (SES)			
Very good and good	69 (95.8%)	401 (95.7%)	>0.05
Middle	2 (2.8%)	17 (4.1%)	
Low	1 (1.4%)	1 (0.2%)	
Smoking before pregnancy			
Never	54 (76.1%)	292 (69.2%)	>0.05
Smoking cessation before pregnancy	10 (14.1%)	79 (18.7%)	
Smoking cessation in this pregnancy	5 (7%)	46 (10.9%)	
Smoking in this pregnancy	2 (2.8%)	5 (1.2%)	
Drinking before pregnancy			
Yes	68 (95.8%)	405 (96%)	>0.05
No	3 (4.2%)	17 (4%)	
Parity			
0	36 (50%)	201 (47.5%)	>0.05
1	27 (37.5%)	151 (35.7%)	
2	6 (8.3%)	46 (10.9%)	
3	2 (2.8%)	19 (4.5%)	
≥4	1 (1.4%)	6 (1.4%)	

**Table 2 children-10-00447-t002:** Adverse pregnancy outcomes.

	**Premature Neonates, Gestation < 37 Weeks** **N = 72**	**Full-Term Neonates, Gestation > 37 Weeks** **N = 423**	** *p* **
**Pregestational diabetes mellitus**			
Yes	4 (5.6%)	6 (1.4%)	*p* < 0.05
No	68 (94.4%)	417 (98.6%)	
**Gestational diabetes mellitus**			
Yes	11 (15.3%)	67 (15.8%)	*p* > 0.05
No	61 (84.7%)	356 (84.2%)	
**Hypothyroidism**			
Yes	35 (48.6%)	170 (40.2%)	*p* > 0.05
No	37 (51.4%)	253 (59.8%)	
**Chronic hypertension**			
Yes	1 (1.4%)	7 (1.7%)	*p* > 0.05
No	71 (98.6%)	416 (98.3%)	
**Gestational hypertension**			
Yes	9 (12.5%)	31 (7.3%)	*p* > 0.05
No	63 (87.5%)	392 (92.7%)	
**Chronic renal disease**			
Yes	0 (0%)	4 (0.9%)	*p* > 0.05
No	72 (100%)	419 (99.1%)	
**Chronic cardiac disease**			
Yes	0 (0%)	5 (1.2%)	*p* > 0.05
No	72 (100%)	418 (98.8%)	
**Chronic pulmonary disease**			
Yes	1 (1.4%)	1 (0.2%)	*p* > 0.05
No	71 (98.6%)	422 (99.8%)	
**Asthma**			
Yes	1 (1.4%)	4 (0.9%)	*p* > 0.05
No	71 (98.6%)	419 (99.1%)	
**Allergy**			
Yes	3 (4.2%)	18 (4.3%)	*p* > 0.05
No	69 (95.8%)	405 (95.7%)	
**Anaemia**			
Yes	5 (6.9%)	42 (9.9%)	*p* > 0.05
No	67 (93.1%)	381 (90.1%)	
**Chronic contagious disease**			
Yes	0 (0%)	2 (0.5%)	*p* > 0.05
No	72 (100%)	421 (99.5%)	
**Depression**			
Yes	0 (0%)	9 (2.1%)	*p* > 0.05
No	72 (100%)	414 (97.9%)	
**Cancer during pregnancy**			
Yes	0 (0%)	2 (0.5%)	*p* > 0.05
No	72 (100%)	421 (99.5%)	
**Combined maternal chronic disease (≥2)**			
Yes	20 (27.8%)	88 (20.8%)	*p* > 0.05
No	52 (72.2%)	335 (79.2%)	
**Progesterone**			
Yes	15 (20.8%)	30 (7.1%)	*p* < 0.001
No	57 (79.2%)	393 (92.9%)	
**Levothyroxine**			
Yes	35 (48.6%)	165 (39.1%)	*p* > 0.05
No	37 (51.4%)	257 (60.9%)	
**Antihypertensive drugs**			
Yes	8 (11.1%)	24 (5.7%)	*p* > 0.05
No	64 (88.9%)	399 (94.3%)	
**Heparin**			
Yes	6 (8.3%)	21 (5%)	*p* > 0.05
No	66 (91.7%)	402 (95%)	

## Data Availability

Not applicable.

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
