# Peer review of "Contributors to Preterm Birth: Data from a Single Polish Perinatal Center"

_children, 2023, doi:10.3390/children10030447_

Round 1

Reviewer 1 Report

In this form, the work is really a short presentation of own results on too small a cause and there is no category of originality, but a short professional presentation with the usual conclusion: no scientific contribution: small sample to test, no new knowledge. The oversimplified way of writing this paper is for presentation at a congress, not for a high IF and Q presentation journal.

Author Response

Dear Reviewer,

Our paper presents the results of work in a tertiary perinatal care facility throughout 1 year period. This is a preliminary study (brief report). Looking into the future, we aim to extend the study and implement data regarding ART-treatment or dividing between the vaginal birth and cesarean sections (which to date are the limitations of our study) in a larger group of patients.

In materials and methods sections there was a mistake in the dates regarding the time spent to collect data – it should be from February 2019 (not November 2019) to February 2020. Incomplete questionnaires were not taken into account. 10% of women refused to take part in the survey. 5% of the questionnaires were incomplete. The rest was enrolled.

Reviewer 2 Report

The manuscript aimed to give a descriptive framework of risk factors associated to preterm labor, considering recent social and environmental challenges that could have change transition to parenthood. 

Some limitation should be observed. First, Introduction section well described the states of art on risk factor for preterm birth. Nevertheless, at lines 61-66, topic moved on new scenarios of parenthood: this change is not well linked with previous one. Furthermore, it is not clear which specific new variables authors evaluate and if some literature already investigated their relationship with prematurity.

In Method section I did not understand if all pregnant women were considered available for research or if some exclusion criteria were used. I only read about twin pregnancy: I have some concern because they are somehow frequent in further population and its exclusion could compromise the generalizability of result. Moreover, if the reason of exclusion is the high association between twin pregnancy and preterm labor, why other similar risk condition were not considered (i.e. FIVET or other ART treatment)?

Another doubt regards the recruitment: all women accepted to partecipate? if not, are they significant different from final sample?

I also did not find info on health practices during pregnancy? did you check the number of visits or exams during pregnancy? because many sociodemographic characteristics reduce the quality of life of women during pregnancy and this contribute to preterm birth: so the relationship could be not direct. This point should be explained or add as limit.

also regarding limit, no one is declared.

I read references and it seemed to be very old. only few articles regards last five years. Pleas update literature or give explanation on possible absence of update data in last years. also the access at website or report is very old (about 2 years ago).

Reference section s should be also corrected according to journal guidelines.

Author Response

Dear Reviewer,

thank you very much for your valuable suggestions. Here we provide a point-by-point response to your review.

“The manuscript aimed to give a descriptive framework of risk factors associated to preterm labor, considering recent social and environmental challenges that could have change transition to parenthood.”

“Some limitation should be observed. First, Introduction section well described the states of art on risk factor for preterm birth. Nevertheless, at lines 61-66, topic moved on new scenarios of parenthood: this change is not well linked with previous one. Furthermore, it is not clear which specific new variables authors evaluate and if some literature already investigated their relationship with prematurity.”

The introduction section has been modified to make it more coherent and cohesive. Also, additional references have been added. A separate limitations sections has been added to the manuscript.

“In Method section I did not understand if all pregnant women were considered available for research or if some exclusion criteria were used. I only read about twin pregnancy: I have some concern because they are somehow frequent in further population and its exclusion could compromise the generalizability of result. Moreover, if the reason of exclusion is the high association between twin pregnancy and preterm labor, why other similar risk condition were not considered (i.e. FIVET or other ART treatment)?”

Referring to ART treatments methods, in Poland questioning about ART treatments might be still a very uncomfortable topic and some mothers do not admit it. Therefore, we did not include ART treatments.

Exclusion criteria were twin pregnancies, inappropriately fulfilled questionnaires and refusal to take part in the study.

“Another doubt regards the recruitment: all women accepted to partecipate? if not, are they significant different from final sample?”

10% of all women refused to participate in the study (mostly due to the current postpartum health condition), 5% of the questionnaires were incomplete. The rest was enrolled.

“I also did not find info on health practices during pregnancy? did you check the number of visits or exams during pregnancy? because many sociodemographic characteristics reduce the quality of life of women during pregnancy and this contribute to preterm birth: so the relationship could be not direct. This point should be explained or add as limit.”

In Poland health insurance is widely common and accessible. All pregnant women are under the medical care of gynecologists and midwifes. Furthermore, regular control appointments during the pregnancy are linked with obtaining certain social benefits, what promotes follow-up visits. In Poland, lack of medical care during the pregnant period or difficulties with access to it are immensely rare. Therefore, we did not include this subject in our studies.

“also regarding limit, no one is declared.”

A paragraph with the limitations of the study has been added to the manuscript.

Limitations of the study are not considering the type of delivery (vaginal birth versus cesarean section) as well as and not considering labors from in vitro fertilization (IVF). The goal of the survey was to collect data about the women’s lifestyle and environmental, social and economic conditions. Therefore, apart from certain concomitant diseases (hypertension, diabetes mellutis or hypothyroidism), study focuses deeply on determining the influence of health behavior of pregnant women on the risk of preterm birth. That is why neither ART treatments nor type of delivery questions were included in the survey. For the same reason we did not ask whether the pregnancy ended up with preterm birth due to iatrogenic reasons (i.e., elective cesarean section).  So far, this is a preliminary study (brief report), and these limitations create a valuable evaluation of the study for the future publications.

“I read references and it seemed to be very old. only few articles regards last five years. Pleas update literature or give explanation on possible absence of update data in last years. also the access at website or report is very old (about 2 years ago)”

Some outdated references have been changed to the most recent ones.

Reviewer 3 Report

It's clear that authors have spent efforts on this work. Here are my comments:

In the introduction:

Line 62 "Numerous women live...." This sentence feels out of context. Could be deleted.

Line 67: "In order to confirm..." would be better if " in order to study" or "further explore".

Line 68: "Pomeranian region" Not everyone is familiar with the Polish Geography, therefore some insight about Poland and where this region located would provide readers with more context.

In Materials and Methods:

Line 87: "multiple pregnancies typically lead..." this is not accurate! You can replace "typically"  with other word to be more precise and scientific.

* Please provide clear: inclusion criteria, exclusion criteria.

* Please mention the limitation of this study.

* Please provide more info about the questionnaire: number of questions, paper vs online , response rate, language  used and whether that would be of any barrier....... etc.

* There are other- important- variables which would alter the results like: natural pregnancies vs IVF pregnancy or any assisted conception, whether delivering moms were in labor or not, or C-sections for maternal reasons?

* Type of delivery is essential; vaginally or  C sections. 

In Discussion:

You have to mention the limitations

In Conclusion:

Generally, the less the better. Try to summarize conclusion in few sentences, one or 2 liners, rather than a full paragraph. It will read better, and have higher impact!

Line: 264: "Women's desire to fulfill themselves ........may lead to postponing the decision" was this a direct question in the survey or rather authors own extrapolation? It doesn't serve conclusion!

Author Response

Dear Reviewer,

thank you very much for the valuable suggestions. Here we provide a point-by-point response to the comments.

It's clear that authors have spent efforts on this work. Here are my comments:

In the introduction:

“Line 62 "Numerous women live...." This sentence feels out of context. Could be deleted.”

Notably, in Poland informal relationships are becoming more and more common, yet are very often sociable unacceptable. This aspect is also not regulated by the Polish law. We deeply believe that this statement is relevant and, therefore, opt to leave it. Moreover, to provide more context we extended the introduction section with psychosocial factors.

“Line 67: "In order to confirm..." would be better if " in order to study" or "further explore".”

Corrected.

“Line 68: "Pomeranian region" Not everyone is familiar with the Polish Geography, therefore some insight about Poland and where this region located would provide readers with more context.”

To give the readers insight about it we deeply evaluated it in the Materials&Methods section just below, providing detailed information about the place the study was performed (Gdańsk, Medical University of Gdańsk, etc.).

In Materials and Methods:

“Line 87: "multiple pregnancies typically lead..." this is not accurate! You can replace "typically" with other word to be more precise and scientific.”

Only singleton pregnancies were considered, as multiple pregnancies statistically more often lead to a preterm birth.

“Please provide clear: inclusion criteria, exclusion criteria.”

Inclusion criteria: singleton pregnancies, Polish-speaking women who have just given birth and obtaining a written consent to take part in the study.

Exclusion criteria were multiple pregnancies (since they statistically more often lead to preterm birth), inappropriately fulfilled questionnaires and refusal to take part in the study.

“Please mention the limitation of this study.”

A paragraph with the limitations of the study has been added to the manuscript.

Limitations of the study are not considering the type of delivery (vaginal birth versus cesarean section) as well as and not considering labors from in vitro fertilization (IVF). The goal of the survey was to collect data about the women’s lifestyle and environmental, social and economic conditions. Therefore, apart from certain concomitant diseases (hypertension, diabetes mellutis or hypothyroidism), study focuses deeply on determining the influence of health behavior of pregnant women on the risk of preterm birth. That is why neither ART treatments nor type of delivery questions were included in the survey. For the same reason we did not ask whether the pregnancy ended up with preterm birth due to iatrogenic reasons (i.e., elective cesarean section).  So far, this is a preliminary study (brief report), and these limitations create a valuable evaluation of the study for the future publications.

Please provide more info about the questionnaire: number of questions, paper vs online , response rate, language  used and whether that would be of any barrier....... etc.”

The survey was anonymous and in Polish language. It consisted of 33 closed single- and multiple-choice questions.  All questionnaires were gathered in written, “paper form”. Only completely and correctly filled questionaries were enrolled to the statistical analysis. Approximately 10% of them were incomplete and were, therefore, were not included in the study - 5% of patients refused to take part in the survey, mostly due to unsatisfying health condition postpartum.

“There are other- important- variables which would alter the results like: natural pregnancies vs IVF pregnancy or any assisted conception, whether delivering moms were in labor or not, or C-sections for maternal reasons?”

Referring to ART treatments methods, in Poland questioning about ART treatments might be still a very uncomfortable topic and some very often do not admit it. Therefore, we did not include ART treatments.

The goal of the survey was to collect data about the women’s lifestyle and environmental, social and economic conditions. It focused on the health behavior of pregnant women, not on the detailed medical interview. That is why neither ART treatments nor type of delivery questions were included in the survey. For the same reason we did not ask whether the pregnancy ended up with preterm birth due to iatrogenic reasons (i.e., elective cesarean section). 

“Type of delivery is essential; vaginally or C sections.” 

It is a limitation of our studies.

In Discussion:

“You have to mention the limitations”

A paragraph with the limitations of the study has been added to the manuscript.

Limitations of the study are not considering the type of delivery (vaginal birth versus cesarean section) as well as and not considering labors from in vitro fertilization (IVF). The goal of the survey was to collect data about the women’s lifestyle and environmental, social and economic conditions. Therefore, apart from certain concomitant diseases (hypertension, diabetes mellutis or hypothyroidism), study focuses deeply on determining the influence of health behavior of pregnant women on the risk of preterm birth. That is why neither ART treatments nor type of delivery questions were included in the survey. For the same reason we did not ask whether the pregnancy ended up with preterm birth due to iatrogenic reasons (i.e., elective cesarean section).  These create a valuable evaluation of the study for the future publications.

In Conclusion:

“Generally, the less the better. Try to summarize conclusion in few sentences, one or 2 liners, rather than a full paragraph. It will read better, and have higher impact!”

The conclusion section has been modified according to suggestions.

Line: 264: "Women's desire to fulfill themselves ........may lead to postponing the decision" was this a direct question in the survey or rather authors own extrapolation? It doesn't serve conclusion!

Regarding this question we are in full agreement, and it will be deleted.

Round 2

Reviewer 1 Report

In this revised form: accept (but in form preliminary study or short communication)

Author Response

Dear Reviewer,

thank you for your opinion. In accordance to you suggestion, in the current form of our article we would like to publish it as a preliminary study (which is a "brief report" type regarding the MDPI article types).

Reviewer 2 Report

I thank authors for the revision that improve the quality of manuscript.

however, I have two further consideration. First, because in Introduction section there is a clear state of art of the role of risk factor for preterm birth, it is not clear why authors chose to further investigation. Given the aim of the study is the assessment of the relevance of these variables for polish population, author could add a sentence explaining that there is a lack of study specific for their country. It also show the strenght of the study and the implication for clinical for polish patients.

second, I appreciated that authors added a limit section. However I think that it should be better described, explaining that these variables are relevant for previous literature and why they were not able to include them in the present research. A consideration of the need of further studies starting from the present results adding info on other variables should be added.

Author Response

Dear Reviewer,

Thank you for your suggestions, we have considered them all deeply.

“However, I have two further consideration. First, because in Introduction section there is a clear state of art of the role of risk factor for preterm birth, it is not clear why authors chose to further investigation. Given the aim of the study is the assessment of the relevance of these variables for polish population, author could add a sentence explaining that there is a lack of study specific for their country. It also show the strenght of the study and the implication for clinical for polish patients.”

In the introduction we have added a sentence explaining that there is a lack of such studies in this field (see lines 69-80). Further investigation reasons are embedded in the limitations section – mainly, due to the increasing number of preterm births in developed countries it is crucial to investigate the iatrogenic reasons responsible for them, what we aim to accomplish in the next study.

“second, I appreciated that authors added a limit section. However I think that it should be better described, explaining that these variables are relevant for previous literature and why they were not able to include them in the present research. A consideration of the need of further studies starting from the present results adding info on other variables should be added.”

The limitations of the study section has been extended and describes the variables in a better, more clear and in-depth way (see linies 315-326).